# Changes in the Microbial Structure of the Root Soil and the Yield of Chinese Baby Cabbage by Chemical Fertilizer Reduction with Bio-Organic Fertilizer Application

Li Jin,[a] Ning Jin,[a] Shuya Wang,[a] Jinwu Li,[a] Xin Meng,[a] Yandong Xie,[a] Yue Wu,[a] Shilei Luo,[a] Jian Lyu,[a,b] Jihua Yu[a,b]

[a]College of Horticulture, Gansu Agricultural University, Lanzhou, Gansu Province, China
[b]Key Laboratory of Crop Science in arid environment of Gansu Province, Lanzhou, Gansu Province, China

**ABSTRACT** Using high-throughput sequencing, this study aimed to explore the response of soil microbial community and Chinese baby cabbage yield to the reduction of chemical fertilizers combined with bio-organic fertilizer. Our experiments consisted of conventional fertilizer (CK), 30% chemical fertilizer reduction + 6,000 kg/ha bio-organic fertilizer (T1), 30% chemical fertilizer reduction + 9,000 kg/ha bio-organic fertilizer (T2), 40% chemical fertilizer reduction + 6,000 kg/ha bio-organic fertilizer (T3), and 40% chemical fertilizer reduction + 9,000 kg/ha bio-organic fertilizer (T4). Compared with CK, soil microbial diversity and richness were higher for all treatments with added bio-organic fertilizer. Principle coordinate analysis (PCoA) showed that the bacterial and fungal communities in T2 and T4 were similar to each other. Redundancy and Spearman's correlation analyses of microbial communities and soil physicochemical properties revealed that reductions in chemical fertilizer rate combined with bio-organic fertilizer had a stronger impact on the fungal than the bacterial community. They also increased the relative abundance of the dominant bacterial and fungal phyla. Chinese baby cabbage yield was relatively higher under the combined bio-organic fertilizer plus reduced chemical fertilizer rate with T2 showing the highest yield. Therefore, this approach is feasible for sustainable agricultural, cost-effective and profitable crop production.

**IMPORTANCE** Chemical fertilizers are commonly used for agriculture, though bio-organic fertilizers may be more efficient. We found that a mixture of bio-organic and moderately reduced chemical fertilizer was more effective than chemical fertilizer alone, as it raised the Chinese baby cabbage yield. Further, the presence of bio-organic fertilizer enhanced overall soil physicochemistry, as well as improved the beneficial bacterial and fungal abundance and diversity. Thus, we found that fertilizer combination sustainably & cost-effectively improves crop & soil quality.

**KEYWORDS** Chinese baby cabbage, bio-organic fertilizer, microbial community, sustainable agriculture, yield

Chinese baby cabbage (*Brassica rapa subsp. pekinensis*) is a subspecies of *Brassica oleracea* in the crucifer family, with tender yellow inner leaves, short growth period, tender and crispy taste, and is loved by consumers (1). To cope with the surge of social demand, the cultivation scale of Chinese baby cabbage has been gradually expanded in recent years. However, traditional agricultural systems usually rely on high inputs of chemical fertilizers to improve soil fertility and crop growth (2). Chemical fertilizer overuse is also accompanied by a reduction in soil nutrient absorption efficiency, soil quality deterioration, greenhouse gas emissions, and aquatic ecosystem eutrophication (3–5).

Under the advocacy of the sustainable development practice known as "zero growth action plan for fertilizer use in China in 2020," organic agriculture based on natural products such as humic acid and natural fertilizer has been implemented to

Address correspondence to Jian Lyu, lvjiangs@126.com, or Jihua Yu, yujihuagg@163.com.

The authors declare no conflict of interest.

improve soil quality. This practice is conducive to agricultural productivity and ecosystem health (6, 7). The application of plant growth-promoting microorganisms to the soil is efficacious in organic agriculture. This approach compensates for the low efficiency of synthetic fertilizer, improves nutrient utilization efficiency and plant growth, reduces fertilizer investment by 50%, and causes no yield loss (8–10). However, organic fertilizer input may also create competition for nutrients between crops and soil microorganisms decomposing the fertilizer (11). To balance this competition, integrated agricultural systems are, nonetheless, supplemented with chemical fertilizers to mitigate nutrient limitation (12). Plant probiotic enrichment in the form of organic substrates, such as bio-organic fertilizers, can help beneficial microorganisms thrive in the soil and improve their function (13–16). Diacono and Montemurro (17) conducted over 20 long-term tests confirming that organic modifiers never lower crop yield. Zhang et al. (18) showed that replacing 30% of the total nitrogen fertilizer (250 kg N/ha) with 3,000 kg/ha compost (equivalent to 60 kg N/ha) improved maize yield, N uptake, and soil fertility, and reduced N loss. Hence, there is a growing demand for bio-based organic fertilizers in agricultural production (19).

Soil microorganisms regulate soil ecosystem functions and are indicators of soil quality (20). They promote nutrient cycling and organic matter transformation, improve plant productivity, and control soilborne diseases (21–23). Microbial abundance, composition, and activity largely determine sustainable agricultural productivity (24). The soil microbiome may be positively or negatively affected by soil disturbances and management practices that alter soil microbiome classification and function (25, 26). Long-term, large-scale chemical fertilizer use leads to the deterioration of soil quality, nutrient imbalances, reductions in soil microbial diversity, loss of microbial structural integrity, and a decrease in sustainable farmland productivity (27). Bio-organic fertilizers contain beneficial microorganisms and organic components that directly or indirectly promote soil nutrient mobilization, have a positive impact on plant health, and improve crop yield (28). Previous studies showed that bio-organic fertilizers stimulate specific microbiota associated with plant disease inhibition such as *Pseudomonas*, *Streptomyces*, *Flavobacterium*, and others (29–31). Therefore, strategies to stimulate the activity of these soilborne microbiota may be particularly effective in helping to suppress plant diseases. Although there are many reports of bio-organic fertilizer applications on crops, each crop has different fertilizer requirements and produces very different results. There are few reports on the benefits produced by the application of bio-organic fertilizers to Chinese baby cabbage, and even fewer studies on the changes in the microbial community of the root system of Chinese baby cabbage after bio-organic fertilizer application.

In this study, we conducted field trials on Chinese baby cabbage under conventional fertilization, 30% chemical fertilizer reduction + 6,000 kg/ha bio-organic fertilizer, 30% chemical fertilizer reduction + 9,000 kg/ha bio-organic fertilizer, 40% chemical fertilizer reduction + 6,000 kg/ha bio-organic fertilizer, and 40% chemical fertilizer reduction + 9,000 kg/ha bio-organic fertilizer. We used directional sequencing of bacterial and fungal communities to analyze the response of the soil microbial community to the foregoing treatments. The aims of this study were to: (i) determine the changes in the physicochemical properties of the soil and the soil microbial community in response to various fertilizer treatments; (ii) evaluate the impact of the different fertilizer treatments on Chinese baby cabbage crop yield; (iii) identify the correlations among between soil microbial community composition, soil properties, and crop yield, (iv) assess the benefits of combining bio-organic and chemical fertilizers, and (v) recommend an optimal fertilization method for the growth of Chinese baby cabbage. The results of this study will optimize crop productivity while minimizing the impact of chemical fertilization on the environment.

## RESULTS

**Soil physicochemical properties under different fertilization regimes.** The physicochemical properties of the soil significantly differed among fertilization treatments (Table 1). Compared with conventional fertilization (CK), the soil total nitrogen (TN), total phosphorus (TP), and total potassium (TK) increased with bio-organic fertilizer

**TABLE 1** Physicochemical properties of soils under different fertilization conditions[a]

| Treatments | TN[b] (g/kg) | TP[c] (g/kg) | TK[d] (g/kg) | SOM[e] (g/kg) | EC[f] (uS/cm) | pH[g] |
|---|---|---|---|---|---|---|
| CK | 0.39 ± 0.02c | 0.83 ± 0.02d | 16.17 ± 0.30b | 33.38 ± 0.58e | 507.67 ± 5.21a | 7.25 ± 0.07c |
| T1 | 0.48 ± 0.01b | 0.94 ± 0.06c | 17.17 ± 0.44ab | 35.78 ± 0.55d | 317.33 ± 2.6d | 7.49 ± 0.01ab |
| T2 | 0.54 ± 0.03a | 1.16 ± 0.03a | 17.42 ± 0.30ab | 38.56 ± 0.91c | 359.00 ± 5.86c | 7.55 ± 0.02a |
| T3 | 0.48 ± 0.01b | 1.06 ± 0.05b | 17.67 ± 0.55a | 43.74 ± 0.33b | 384.00 ± 4.04b | 7.43 ± 0.12ab |
| T4 | 0.50 ± 0.01ab | 1.11 ± 0.04ab | 17.83 ± 0.51a | 49.43 ± 0.77a | 397.00 ± 5.87b | 7.48 ± 0.08ab |

[a]Values (mean ± standard deviation) followed by different lowercase letters within a column for the same factor indicate significant differences by Tukey test ($P < 0.05$).
[b]TN, total nitrogen.
[c]TP, total phosphorus.
[d]TK, total potassium.
[e]SOM, soil organic matter.
[f]EC, soil electrical conductivity.
[g]pH, soil pH.

application rate. TN and TP were highest for the soil under 30% chemical fertilizer reduction + 9,000 kg/ha bio-organic fertilizer (T2), and were 23.08% and 39.76% greater than those under CK. Compared with CK, TP under 40% chemical fertilizer reduction + 6,000 kg/ha bio-organic fertilizer (T3) and 40% chemical fertilizer reduction + 9,000 kg/ha bio-organic fertilizer (T4) were 7.73% and 10.27% higher, and the differences were significant ($P < 0.05$). The soil organic matter (SOM) content was lowest for CK and was significantly lower than the SOM under all other treatments ($P < 0.05$). The SOM content increased sequentially from 30% chemical fertilizer reduction + 6,000 kg/ha bio-organic fertilizer (T1) to T4 ($P < 0.05$). All values were significantly higher than that of CK ($P < 0.05$). The SOM content was highest under T4 and 48.08% greater than the SOM content under CK. The soil EC was highest under CK and was significantly higher than the soil EC under all other treatments ($P < 0.05$). The soil EC under T1 through T4 were 37.49%, 29.28%, 24.36%, and 21.80% lower than the soil EC under CK, respectively, and the differences were significant ($P < 0.05$). The soil pH was highest under T2 and was 4.14% higher than the soil pH under CK. The difference was significant ($P < 0.05$). The soil pH under all other treatments did not significantly differ from that under T2 but were significantly higher than that under CK ($P < 0.05$).

**Changes in soil microbial $\alpha$-diversity under different fertilization regimes.** We measured the Shannon, Simpson, and Chao indices of bacteria (Fig. 1 A to C) and fungi (Fig. 1 D to F) to evaluate the $\alpha$-diversity of the soil microbial community in the Chinese baby cabbage root soil under different fertilization treatments. The Shannon index is directly proportional to the diversity while the Simpson index is inversely proportional to it. The Chao index is directly proportional to the microbial abundance. Figure 1A shows that, after the various fertilization treatments, the Shannon index was 0.63% and 0.78% higher for the soil bacteria under T1 and T3, respectively, than it was for those under CK, and the differences were significant ($P < 0.05$). There was no significant difference between T4 and CK in terms of the Shannon index ($P < 0.05$). However, the Shannon index was 1.10% lower under T2 than CK and the difference was significant ($P < 0.05$). There were no significant differences among fertilization treatments in terms of their bacterial Simpson indices (Fig. 1B). The bacterial Chao index was highest under T4 and was 34.20% higher than that under CK. However, there were no significant differences among the other treatments and CK in terms of the bacterial Chao index (Fig. 1C). The fungal Shannon indices under T1 through T4 were 10.66, 14.42, 8.46, and 14.73% higher, respectively, than that under CK, and the differences were significant (Fig. 1D) ($P < 0.05$). The fungal Simpson indices under T1 through T4 were 44.53, 53.13, 34.38, and 50.78% lower, respectively, than that under CK, and the differences were significant (Fig. 1E) ($P < 0.05$). The fungal Chao indices under T2 and T4 were 23.40% and 23.49% higher, respectively, than that under CK, and the differences were significant (Fig. 1F).

**Changes in soil microbial $\alpha$-diversity under different fertilization regimes.** We applied a principal coordinate analysis (PCoA) based on the Bray-Curtis algorithm to

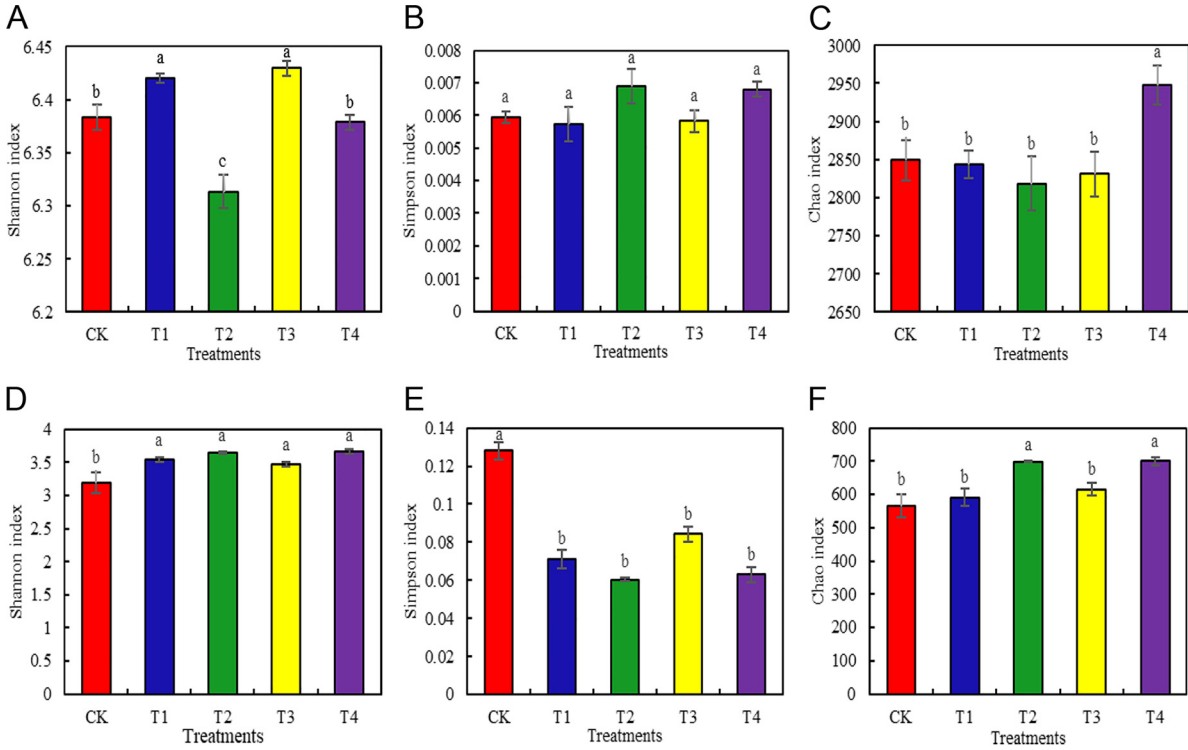

**FIG 1** The $\alpha$-diversity of bacteria (A) to (C) and fungi (D) to (F) in the root soil of Chinese baby cabbage under different fertilization treatments. Different letters over the plots indicate significant differences ($P < 0.05$).

analyze overall structural changes in the bacterial and fungal microbial communities and determine the impact of reducing chemical fertilizers and adding bio-organic fertilizers on their structure. The fertilization regimes significantly altered soil bacterial and fungal community structure in the Chinese baby cabbage root soil. The PCoA assigned the bacterial communities to 2 groups, namely, (CK, T1, and T3) and (T2 and T4) (Fig. 2A). The T2 and T4 bacterial community structures were clustered together and were remote from those of the other treatment groups (Fig. 2A). Hence, the soil bacterial community structure significantly changed with increasing bio-organic fertilizer application rate. The PCoA assigned the fungal communities to 3 groups, namely, (CK), (T1 and T3), and (T2 and T4) (Fig. 2B). The T2 and T4 fungal community structures were clustered together and were remote from that of CK (Fig. 2B). There was a certain distance between T1 and T3, but both were nonetheless in the same quadrant. Thus, the change in the fungal community structure of the Chinese baby cabbage root soil depended on whether biological organic fertilizer was used, and the quantity added. PERMANOVA revealed that each fertilization treatment significantly affected the bacterial ($F = 1.64$, $P = 0.01$) and fungal ($F = 6.66$, $P = 0.01$) community structures.

**Associations between soil microbial communities and environmental factors.** A redundancy analysis (RDA) was used to evaluate the influences of environmental factors on the soil microbial community composition in Chinese baby cabbage root soil (Fig. 3). The changes in the soil properties in response to the different fertilization treatments strongly affected the soil bacterial and fungal community structures. In Fig. 3A, the first 2 axes of RDA explain 16.29% and 3.34% of the total variation in the soil bacterial data, respectively. In Fig. 3C, the first 2 axes of the RDA explain 80.34% and 1.22% of the total variation in the soil fungal data, respectively. The first axis in Fig. 3A was positively correlated with soil TN, TK, TP, pH, and SOM and negatively correlated with soil EC. The second axis was positively correlated with soil EC, TP, and pH and negatively correlated with soil TN, TK, and SOM. We also used Spearman's rank correlation to evaluate the relationships among the abundant bacterial phyla and soil physicochemistry (Fig. 3B). *Bacteroidota* were significantly, positively correlated with

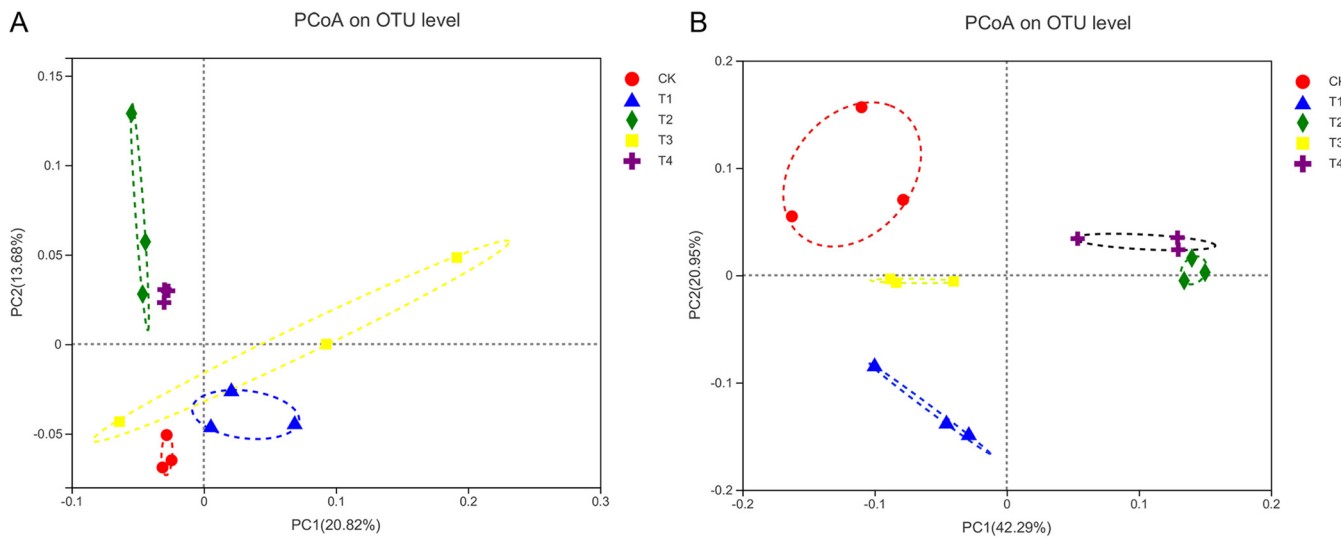

**FIG 2** Principal coordinate analysis (PCoA) of the bacteria (A) and fungi (B) communities based on the Bray–Curtis distances.

soil TN and TK, *SAR324_cladeMarine_group_B* and *Firmicutes* were significantly, negatively correlated with soil EC, *Acidobacteriota* were significantly, negatively correlated with soil TN and pH, and *Methylomirabilota* and *Latescibacterota* were significantly, negatively correlated with SOM ($P < 0.05$). The first axis in Fig. 3B was positively correlated with soil TN, TK, TP, pH, and SOM and negatively correlated with soil EC. The second axis was positively correlated with soil EC and SOM and negatively correlated with soil TN, TK, TP, and pH. Spearman's rank correlation between the soil fungal community and physicochemistry disclosed that *Zoopagomycota*, *unclassified_k__Fungi*, and *Mortierellomycota* were significantly, positively correlated with soil TP, SOM, and TN, respectively ($P < 0.05$), and *Ascomycota* were significantly, positively correlated with soil TK, SOM, and TN ($P < 0.05$), and significantly, positively correlated with soil pH ($P < 0.01$). *Mortierellomycota*, *Kickxellomycota*, and *Blastocladiomycota* were significantly, negatively correlated with soil EC, while *Olpidiomycota* were significantly, negatively correlated with SOM and soil pH ($P < 0.05$).

**Relative abundance of major bacterial and fungal taxa.** The main bacterial phyla were *Actinobacteriota*, *Proteobacteria*, *Chloroflexi*, and *Acidobacteriota* (Fig. 4A), and their average relative abundances were 35.14, 19.83, 16.39, and 10.56%, respectively. The subdominant bacterial phyla were *Gemmatimonadota*, *Bacteroidota*, *Myxococcota*, and *Firmicutes*. T1 through T4 had higher *Actinobacteria* richness than that of CK. T2 and T4 had the highest and second highest *Actinobacteriota* abundance. *Actinobacteriota* abundance was higher under T1 than T3. T3 and T4 had higher *Proteobacteria* abundance than CK. However, *Proteobacteria* abundance did not significantly ($P > 0.05$) differ between T1 and T2. Compared with CK, *Chloroflexi*, *Acidobacteriota*, and *Gemmatimonadota* had higher abundance, while *Bacteroidota* and *Firmicutes* had lower abundance under T1 through T4. The main fungal taxa were *Ascomycota*, *Olpidiomycota*, *Mortierellomycota*, *Basidiomycota*, and *unclassified_k__Fungi* (Fig. 4B). T1 through T4 had higher *Ascomycota* abundance than CK. T2 had the highest *Ascomycota* abundance but relatively low *Olpidiomycota* abundance. T1 through T4 had relatively high *Mortierellomycota* abundance.

**Effect of different fertilization regimes on Chinese baby cabbage yield.** We measured Chinese baby cabbage yield in 2019 and 2020, and calculated the average for both years. Compared with CK, T1 through T4 significantly increased Chinese baby cabbage yield by 6.39, 8.89, 3.94, and 5.14%, respectively ($P < 0.05$). The yield was highest under T2 but did not significantly differ from that under T1 (Fig. 5).

## DISCUSSION

Widespread use of chemical fertilizers has led to losses of soil quality and organic

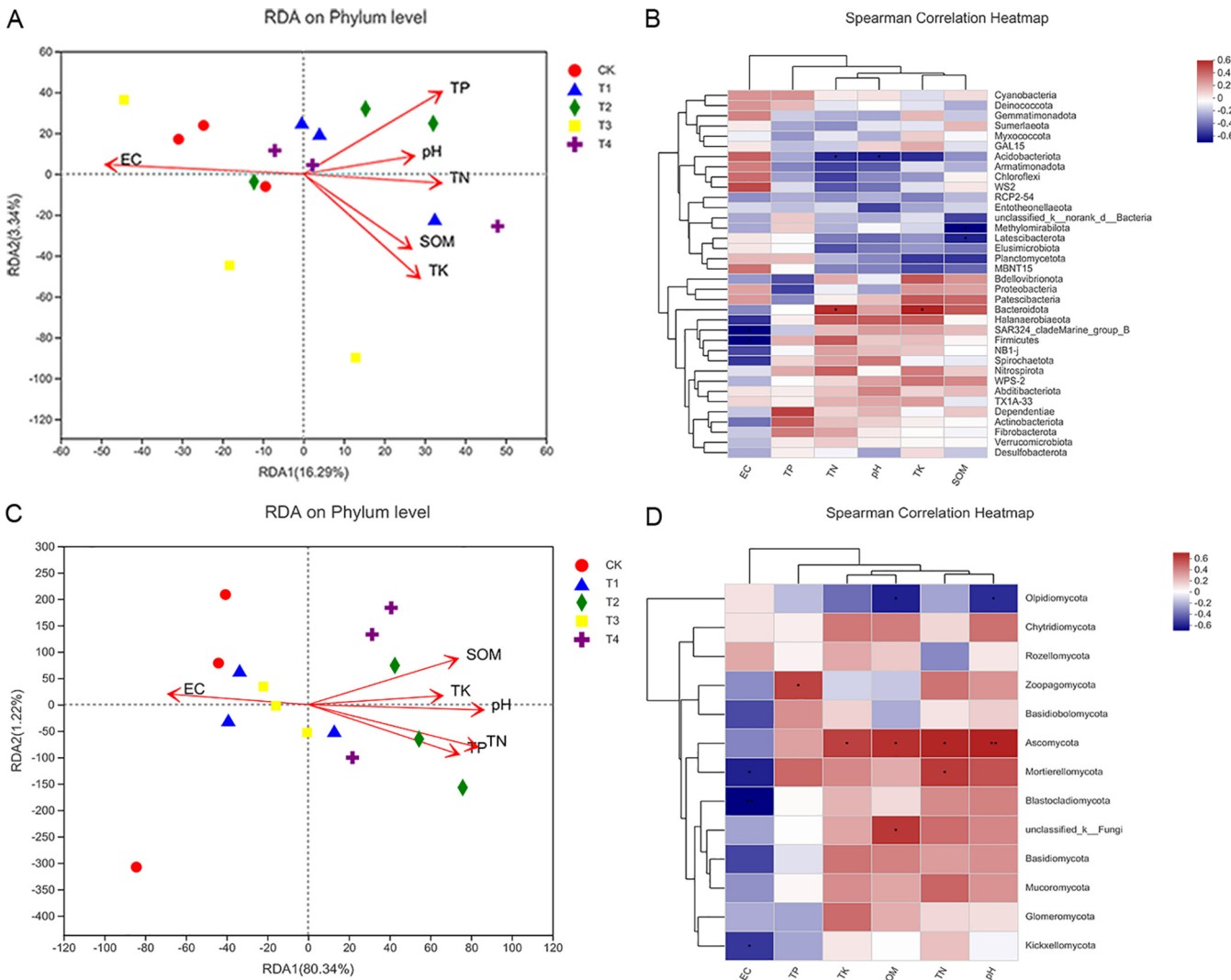

**FIG 3** Redundancy analysis (RDA) and Spesrman's rank correlation heatmap ($P < 0.05$ *, $P < 0.01$**), used to study the correlation between the microbial communities of bacteria (A) and (B) and fungi (C) and (D) and soil physical and chemical properties. TP, total phosphorus; TK, total potassium; TN, total nitrogen; pH, soil pH; EC, soil electrical conductivity; SOM, soil organic matter.

carbon content (32, 33). Therefore, fertilization must be optimized to ensure sustainable food production. Compared with conventional soil cultivation, organic soil cultivation enhances microbial functional diversity (34). Long-term application of organic fertilizers or organic-inorganic compound fertilizers may increase microbial biomass and enzyme activity and improve SOM quantity and quality (35). In our study, compared with conventional fertilization, all four chemical fertilizer reduction and combined with bio-organic fertilizer application treatments significantly increased soil TP, TN, TK, pH, and SOM. Furthermore, both TK and SOM increased with bioorganic fertilizer application rate (Table 1). Therefore, the combined application of organic and chemical fertilizers increases soil fertility and supports sustainable productivity more effectively than chemical fertilizer application alone (36, 37). The addition of *Bacillus subtilis* and *Pseudomonas stutzeri* to the organic fertilizer further increased soil extracellular enzyme activity and nutrient content (38). TP, TN, and pH were at their maxima under T2 (Table 1). For this reason, even moderate reductions in chemical fertilizer application rate followed by bio-organic fertilizer treatment effectively increase soil nutrient content and improve soil quality. By contrast, excessive reductions in chemical fertilizer application may decrease soil nutrient levels even when sufficient bio-organic fertilizer is used.

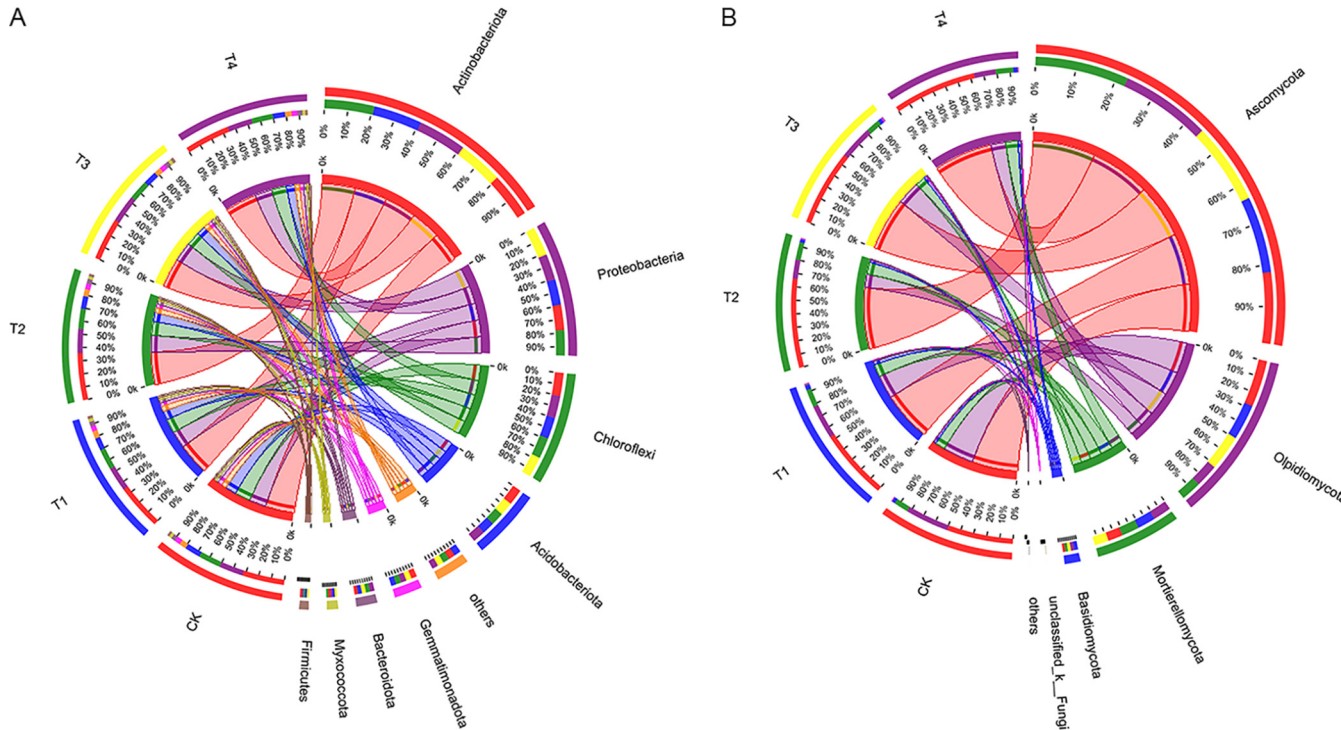

**FIG 4** The relative abundance of major taxonomic groups at the phylum level (Others incorporated < 0.01) for bacteria (A) and fungi (B). The data were visualized by Circos. The width of the bars from each phylum indicated the relative abundance of the phylum.

Several studies evaluated the effects of fertilization on soil microbes. Organic fertilizers, such as manure might have a positive effect on the soil microbial community. Compared with chemical or no fertilization, organic fertilization improves the resistance of the soil microbial community to perturbations (39–41). Our results showed that compared with conventional fertilization, T1 and T3 significantly increased bacterial diversity while T4 significantly increased bacterial richness (Fig. 1A to C). Previous studies have also shown that under chemical fertilizer reduction conditions, organic supplementation can significantly alter the soil bacterial community structure, while also significantly increasing bacterial species richness, chao1, and Shannon index values (42, 43). This may be because organic fertilizers can not only provide greater diversity of microbial active substrates than inorganic fertilizers, but also directly introduce naturally occurring microorganisms into the soil (44). In our experiments, organic fertil-

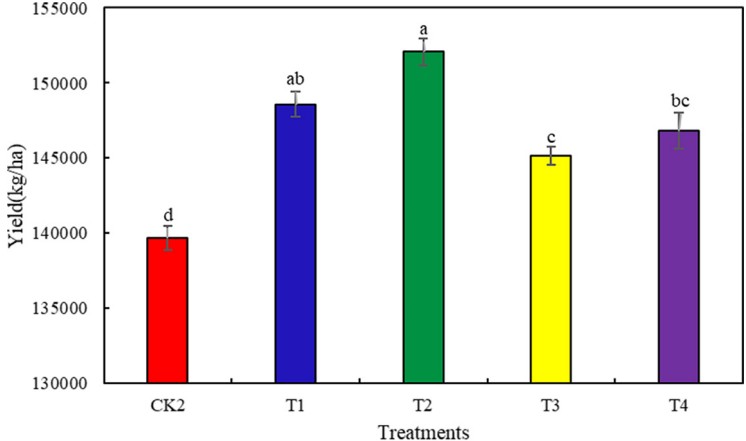

**FIG 5** Average yield of Chinese baby cabbage under different fertilization systems in 2019–2020. Different letters over the plots indicate significant differences ($P < 0.05$).

izers were supplemented with additional beneficial microorganisms, which beneficially increased the diversity and richness of the bacterial community. In addition, the novelty of our study was to compare the effects of different ratios of chemical fertilizers and bio-organic fertilizers on the diversity and richness of microbial communities. From our results, it is evident that different ratios of chemical fertilizers and bio-organic fertilizers have varying effects on the diversity and richness of bacterial communities. Therefore, optimized fertilizer ratios should be selected during the cultivation of Chinese baby cabbage. Compared with conventional fertilization, organic fertilization had a stronger effect on soil fungal communities (45, 46). All treatments with bio-organic fertilizer increased soil fungal community diversity and richness. However, T2 and T4 were the most effective (Fig. 1D to F). That is, the effective enhancement of soil microbial diversity and richness may be a combined effect of Bioorganic and chemical fertilizer applications (47).

The soil microbial community structure significantly differed between the bio-organic fertilizer and other treatments. The PCoA showed that the bacterial and fungal community structures under T2 and T4, and those under T1 and T3 were similar. In both cases, they were separated by CK (Fig. 2A and B). These discoveries were consistent with previous observations that soil microbial community structures differ in response to organic and inorganic fertilizers (48, 49). Moreover, fertilization regimes more strongly influence fungal than bacterial community structure (50, 51).

A combination of PCoA and RDA demonstrated that fertilization may modulate the impact of environmental factors on fungal community structure. The variance between fungal community structure and environmental factors was 81.56% (Fig. 3C). By contrast, the variance between bacterial community structure and environmental factors was only 19.63% (Fig. 3A). Fertilizer application increases the abundance of oligotrophic organisms such as *Bacteroidota*, which are significantly positively correlated with soil TN and TK (Fig. 3B). Oligotrophic organisms tend toward K-selection (52). A high relative abundance of K-strategists promotes microbial community resistance and stability (53). Soil pH is positively correlated with most bacterial communities but negatively correlated with *Acidobacteriota* (Fig. 3B). In this study, the soil pH was near neutrality and organic fertilizer application prevented soil acidification. In this manner, it buffered the potential effects of soil acidity on bacterial richness and diversity (54). However, fungal communities are comparatively more strongly affected by the type of fertilization system. Fungal richness is positively correlated with soil N availability (55). *Ascomycota* are the most abundant fungi in the Chinese baby cabbage root soil and are significantly, positively correlated with TN (Fig. 3D). Hence, the fungal community is closely related to the soil nitrogen fertilizer application rate. SOM also plays an important role in fungal community composition (56, 57). Most dominant fungal phyla such as *Ascomycota*, *Zoopagomycota*, *unclassified_k__Fungi*, and *Mortierellomycota* are positively correlated with SOM. Increases in EC are the result of increases in salinity caused by continuous soil cropping. Soil salinity has negative effects on soil microbial communities (58). The application of bio-organic fertilizer can reduce EC. Most dominant fungal phyla are negatively correlated with this parameter (Fig. 3D).

In all fertilization treatments, *Actinobacteria*, *Proteobacteria*, *Chloroflexi*, and *Acidobacteria* were the dominant bacterial phyla (Fig. 4A). This observation was consistent with those of previous studies (59, 60). *Actinobacteriota* comprise a group of co-nutrients suitable for plant growth in high-C environments (61). Bio-organic fertilizers promote *Actinobacteriota* proliferation because they create nutrient- and carbon-rich environments and their abundance significantly increased in response to bio-organic fertilizer treatment and was highest under T2. Composting is beneficial to the growth of *Proteobacteria* and *Firmicutes* (62). In this study, bio-organic fertilizer treatment also increased *Proteobacteria* and *Firmicutes* abundance. This change in abundance may be due to the presence of *B. subtilis* (*Firmicutes*) and *P. stutzeri* (*Proteobacteria*) in the bio-organic fertilizers used in this study. *Bacillus* spp. and *Pseudomonas* spp. are widely used bacterial and fungal biocontrol agents, and are used to control soilborne plant diseases (63, 64). Cooperation between *Bacillus* spp. and *Pseudomonas* spp. can induce positive interactions and generate multispecies biofilms at the root-microbiome

interface, which can potentially trigger microbial root colonization and resultant plant disease resistance (65–68). In addition, in this study, only *B. subtilis* and *P. stutzeri* were included in the bio- organic fertilizer formula. The enrichment of other potentially beneficial microorganisms in the root soil of Chinese baby cabbages may be due to the inhibition of soilborne pathogens and rebalancing of the microbial community structure. Therefore, the strategy of reducing chemical fertilizer and applying bio-organic fertilizer for changing plant soil microbial community composition has potential for controlling soilborne diseases and improving plant growth. *Chloroflexi* include numerous acid-producing bacteria that anaerobically digest food waste and produce methane gas (69). Long-term chemical fertilizer application may acidify the soil (70), thereby reducing soil microbial diversity and community structure (71). In this study, *Chloroflexi* abundance was higher under the conventional fertilization treatment. The addition of bio-organic fertilizer increased the soil pH (Table 1) and reduced *Chloroflexi* abundance. *Acidobacteria* are slow-growing oligotrophs that flourish under low-nutrient conditions. Their growth is inhibited by N input (72). Bio-organic fertilizer addition increased soil nutrient levels and inhibited *Acidobacteria* growth. Nevertheless, further research is required to elucidate the mechanism(s) by which bio-organic fertilizer addition increases bacterial abundance.

*Ascomycota* were the dominant fungi in this study (Fig. 4B). These saprophytes can thrive in arid environments, have strong environmental adaptability, degrade organic substrates, and are major decomposers of SOM containing cellulose, lignin, and pectin (73). The combination of bio-organic fertilizer plus chemical fertilizer reduction increased *Ascomycota* abundance. *Ascomycota* richness was highest under T2. *Mortierellomycota* are saprophytic and ubiquitous. They can dissolve phosphorus, increase crop yield, and form symbiotic relationships with plants (74, 75). In this study, *Mortierellomycota* abundance increased in response to bio-organic fertilizer treatment and was significantly positively correlated with TP, SOM, and TN (Fig. 3D). Therefore, improvements in the soil nutrients in Chinese baby cabbage root soil may be associated with higher relative *Mortierellomycota* abundance.

The production and integration of new bio-organic fertilizers with beneficial microorganisms and mature compost is an important way to increase crop yields and/or strengthen the control of soilborne diseases (76, 77) . It is feasible but difficult to reduce fertilizer application without causing a loss of productivity (8). The results of this experiment confirmed our hypothesis that bio-organic fertilizer significantly increased Chinese baby cabbage yield compared with chemical fertilizer. T2 (30% chemical fertilizer reduction + 9,000 kg/ha bio-organic fertilizer) realized the highest crop yield of all treatments. Ye et al. (78) showed that a combination of chemical fertilizer reduction and bio-organic fertilizer application could improve crop yield and quality.

These results provide a foundation for understanding how the addition of bioorganic fertilizers caused soil physicochemical improvement and changes in soil microbial diversity and richness, which, in turn, enhanced Chinese baby cabbage yield (Fig. 6).

**Conclusions.** This study examined the effects of various fertilization regimes on soil microbial diversity and community structure in the soil of Chinese baby cabbage grown on the Gansu Plateau. The potential value of bio-organic fertilization in Chinese baby cabbage cultivation was assessed in terms of microbial structure and crop yield. Reduction of chemical fertilizers and addition of bio-organic fertilizers improved soil physicochemistry, markedly altered the soil bacterial and fungal communities, increased their diversity and abundance, and improved the soil microenvironment. Compared with conventional fertilization, the combination of bio-organic fertilizer and reduced chemical fertilizer created a richer, more diverse microenvironment and realized higher Chinese baby cabbage yield. Overall, the T2 fertilization formulation (30% chemical fertilizer reduction + 9,000 kg/ha bio-organic fertilizer) was more beneficial for soil microbial community structure regulation in the root soil of Chinese baby cabbages, which further promoted plant growth and development. Hence, this novel approach is relatively more efficient and desirable in terms of sustainable and high crop yield. Future research should investigate the changes that occur in the

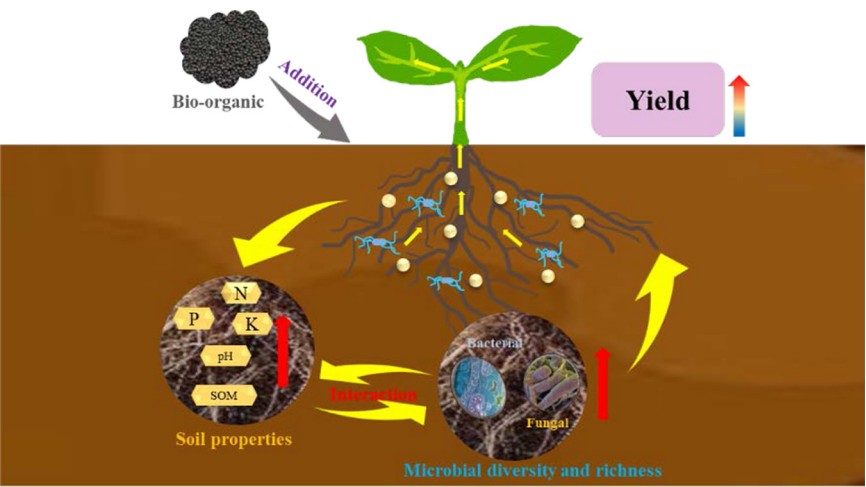

**FIG 6** A model of soil physical and chemical properties, microbial community, and yield changes after adding bio-organic fertilizer.

microbial profile of the soil and the effects of these microorganisms on the yield and quality of Chinese baby cabbage subjected to long-term bio-organic fertilization.

## MATERIALS AND METHODS

**Bio-organic fertilizer preparation.** Bio-organic fertilizer was provided by Gansu Luneng Agricultural Science and Technology Co., Ltd., and was prepared as follows:

*B. subtilis* and *P. stutzeri* were provided by the Microbial Strain Conservation Center of Gansu Province and composed the microorganisms in the biological product. *B. subtilis* and *P. stutzeri* were added to the sterilized beef extract peptone medium (3.0 g beef extract, 10.0 g peptone, 5.0 g sodium chloride, 1000 mL water, 20.0 g agar; pH 7.4 to 7.6, 121°C high-pressure steam sterilized for 20 min) in a 500 mL shaking flask, and cultured at 30°C and 200 r/min for 24 h. The suspension of *B. subtilis* and *P. stutzeri* was then prepared into a compound bacterial solution at a 1:1 ratio. This complex bacterial solution was directly used to prepare biological products as described below.

The organic fertilizer used for biological products was composed of sheep manure and humic acid (formed from activated weathered coal) compost. Sheep manure and humic acid (4:1, wt/wt) were mixed evenly and stacked into strip ridges with a width of 1.5 to 5.0 m and height of 0.8 to 2.5 m for aerobic fermentation. The fermentation temperature was maintained between 40 and 60°C, and the piles were turned over every 4 to 6 days until the end of fermentation. The content of organic matter, humic acid, and water in the fermented organic fertilizer were 40%, 25%, and 30%, respectively, and contained 2.1% nitrogen, 0.7% $P_2O_5$, 1.2% $K_2O$, and pH = 6.8.

Bio-organic fertilizer was prepared by adding compound bacterial solution (inoculation amount: 20%) to the above fermented organic fertilizer and fermenting again for 5 to 6 days. After fermentation, the density of *B. subtilis* and *P. stutzeri* in the bio-organic fertilizer was $0.2 \times 10^8$ CFU/g.

**Experiment design and sampling.** The experiment was conducted in Dachaigou Town (37°09′N, 102°99′E), Tianzhu County, Gansu Province between July 2019 and September 2020. This region has a continental plateau monsoon climate, altitude of ~2,630 m above sea level, annual average temperature of −2°C, annual average precipitation of 400 to 450 mm, and average annual sunshine hours of 2,500 to 2,700 h. The test site had a flat terrain and loam soil with medium, uniform fertility.

The Chinese baby cabbage variety was "DeQin Golden Queen" produced by Beijing De Runcheng Agricultural Science and Technology Development Co. Ltd. (Beijing, China). The experiment was conducted over a 2 year period. In 2019, the Chinese baby cabbage was planted on July 10 and harvested in September. In 2020, the Chinese baby cabbage was planted in July and harvested in September. The fertilizers were urea (N ≥ 46%), calcium superphosphate ($P_2O_5$ ≥ 16%), potassium sulfate ($K_2O$ ≥ 52%), and bio-organic fertilizer. The calcium superphosphate and bio-organic fertilizer were applied as basic fertilizer. Thirty percent of the urea and potassium sulfate was applied as basal fertilizer, another 30% was applied at the rosette stage, and the remaining 40% was applied at early heading.

The experimental design comprised of CK, T1, T2, T3, and T4. Each plot was 0.008 ha in area and the cultivation mode was film-mulching ridge-furrow. The cultivation density was 59,550 plants/ha. The ridge and furrow widths were 50 cm and 40 cm, respectively, and the planting and row distances were 33 cm and 30 cm, respectively. Field management was consistent with local traditional cultivation. The fertilization rates under each treatment are listed in Table 2.

The first crop of Chinese baby cabbage was harvested on September 16, 2019, and its yield was measured. The second crop of Chinese baby cabbage was harvested on September 11, 2020, to determine its yield, and soil samples were collected. A 5-point random sampling method was adopted for each plot of land, and the area of each sampling point was 1 m × 1 m (79–82), and there were 6 plants in total where all of them were collected. A total of 30 plant roots were selected for soil sampling. The

**TABLE 2** Fertilization rates for each treatment

| Treatments | Total amt of fertilization over the entire growth period (kg/ha) | | | |
|---|---|---|---|---|
| | N | P | K | Bio-organic fertilizer |
| CK | 566.3 | 321.4 | 129.1 | 0.0 |
| T1 | 348.3 | 166.6 | 196.9 | 6000 |
| T2 | 348.3 | 166.6 | 196.9 | 9000 |
| T3 | 298.5 | 142.8 | 168.7 | 6000 |
| T4 | 298.5 | 142.8 | 168.7 | 9000 |

specific sampling details were: surface soil was removed with a shovel from 0 to 5 cm depth. The roots of the plants were dug out, leaving them intact as much as possible, and then any large clods of soil around the roots were shook off. The soil near the root systems was brushed with a brush, sealed in bags, placed in ice boxes, and taken to the laboratory. The soil samples were divided into 2 parts: One was air-dried and used in the physicochemical property determinations, and the other was sieved through a 2-mm screen and stored at $-80°C$ for DNA extraction.

**Analysis of soil physical and chemical properties.** Soil physical and chemical properties were determined using the method of Lyu et al. (83). Soil pH and EC were measured with a pH Meter (PHS-3E; Shanghai Jingke) and an EC Meter (DSJ-308Al Shanghai Jingke) (1:5 soil: water [w: v]), respectively. The soil organic matter content was measured by the potassium dichromate method. Total N, P, and K were determined by the Kjeldahl, molybdenum antimony colorimetry, and flame spectrophotometry (FP6410) methods, respectively.

**DNA extraction and PCR amplification.** Microbial genomic DNA extraction from soil samples was performed with the E.Z.N.A. Soil DNA Kit (Omega Bio-tek). DNA quality assays were performed on 1% agarose gels. DNA concentration and purity were determined with a NanoDrop 2000 UV-vis spectrophotometer (Thermo Fisher Scientific) for determination. The V3-V4 region of the bacterial 16S rRNA gene was amplified with primers s338F (5′-ACTCCTACGGGAGGCAGCAG-3′) and 806R (5′-GGACTACHVGGGTW TCTAAT-3′). The fungal ITS2 region was amplified with primers ITS1F (5′-CTTGGTCATTTAGAGGAAGTAA-3′) and ITS2R (5′-GCTGCGTTCTTCATCGATGC-3′). PCR was performed using an ABI GeneAmp 9700 PCR thermocycler (Applied Biosystems). The PCR amplification process was as follows: initial denaturation at 95°C for 3 min, denaturation at 95°C for 27 cycles of 30 s, annealing at 55°C for 30 s, extension at 72°C for 45 s, one-time extension at 72°C for 10 min, and termination at 4°C. A total volume of 20 $\mu$L of PCR mixture included 4 $\mu$L of 5× TransStart FastPfu buffer, 2 $\mu$L of 2.5 mM dNTPs, 0.8 $\mu$L of 5 $\mu$M (each) forward and reverse primers, 0.4 $\mu$L of TransStart FastPfu DNA polymerase, 10 ng of template DNA, and sufficient ddH$_2$O. PCR Reactions were performed in triplicate. PCR products were electrophoresed on a 2% agarose gel, then purified using the AxyPrep DNA Gel Extraction Kit (Axygen Biosciences) and quantified using a Quantus fluorometer (Promega).

**Illumina MiSeq sequencing.** Purified amplicons were pooled in equimolar quantities and paired-end sequenced (2 × 300) according to standard protocols on an Illumina MiSeq platform (Illumina) by Majorbio Bio-Pharm Technology Co. Ltd. The registration numbers for the sequencing data are: SRP359012.

**Processing of sequencing data.** The raw 16S rRNA and ITS gene sequencing reads were demultiplexed, quality-filtered with Trimmomatic (https://github.com/usadellab/Trimmomatic), and merged by FLASH (https://sourceforge.net/projects/flashpage/files/) using the following criteria: (i) The 300-bp reads were truncated at any site with an average quality score <20 over a 50-bp sliding window. Truncated reads <50 bp and reads containing ambiguous characters were discarded. (ii) Only overlapping sequences >10 bp were assembled. The maximum mismatch ratio of the overlap region was 0.2. Reads that could not be assembled were discarded. (iii) Samples were distinguished by their barcodes and primers and the sequence direction was adjusted. Exact barcode matching was used, and 2 nucleotide mismatches were permitted in the primer matching.

The clean tags with sequence similarity greater than 97% were designated as an OTU using UPARSE v. 7.1 (http://drive5.com/uparse/), and then clustering was performed. The taxonomy of each OTU representative sequence (0.7 confidence threshold) was analyzed using the RDP classifier (http://sourceforge.net/projects/rdp-classifier/) against the fungal ITS (Unite 8.0) and bacterial 16S rRNA (Silva SSU128) databases.

**Determination of yield.** On September 16, 2019 and September 11, 2020, we used a 5-point sampling method for yield determination. We artificially harvested in each sample point with an area of 1 m × 1 m, and a total of 30 plant samples were harvested for each treatment to obtain the fresh weight. The yield per hectare was given by yield = (n * W)/30, where W is the weight of 30 Chinese baby cabbages, and n is the number of Chinese baby cabbages per hectare.

**Statistical analysis.** Microbiological data analyses were conducted in R v. 3.5.2 (84). The Shannon, Simpson, and Chao indices were calculated with QIIME (https://qiime2.org). A PCoA was performed based on the Bray-Curtis distance, and it evaluated the similarities of the microbial community composition among samples. A RDA and Spearman's rank correlation heat map analysis were used to study the relationships between soil physicochemical properties and soil microbial communities. Permutational multivariate ANOVA (PerMANOVA) was used to assess the effects of different fertilization regimes on soil microbial communities. PCoA, RDA, and PerMANOVA was implemented using the functions in the R vegan package (85). The Circos graph was plotted with Circos-0.67-7 (http://circos.ca/) (86). SPSS v. 21.0

(SPSS Inc.) was used to perform basic statistical tests such as one-way analysis of variance (ANOVA) and Pearson's correlation analysis. Significant differences among treatments were indicated by $P < 0.05$ or other $P$ values.

**Data availability.** The data sets generated during and/or analyzed during the current study are available from the corresponding author on reasonable request. Sequencing data are stored in the NCBI database, access link: https://www.ncbi.nlm.nih.gov/sra/?term=SRP359012.

## ACKNOWLEDGMENTS

This research was funded by the Education science and technology innovation project of Gansu Province (GSSYLXM-02), the Special project of central government guiding local science and technology development (ZCYD-2021-07), Gansu people's livelihood science and technology project (20CX9NA099), the Fuxi Young Talents Fund of Gansu Agricultural University (GAUfx-04Y03), and Gansu Top Leading Talent Plan (GSBJLJ-2021-14).

This article thanks the experimental field provided by Tianzhu experimental station, as well as the editors and several reviewers for their valuable comments on this article.

We declare that there are no conflicts of interest.

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
