## [Reviewer comments · Microbiology Spectrum]

Microbiology Spectrum

Changes in the microbial structure of the root soil and the yield of Chinese baby cabbage by chemical fertilizer reduction with bio-organic fertilizer application

Li Jin, Ning Jin, Shuya Wang, Jinwu Li, Xin Meng, Yandong Xie, Yue Wu, Shilei Luo, Jian Lyu, and Jihua Yu

Corresponding Author(s): Jian Lyu, Gansu Agricultural University

Review Timeline:

Submission Date:	April 5, 2022
Editorial Decision:	July 15, 2022
Revision Received:	August 29, 2022
Editorial Decision:	October 18, 2022
Revision Received:	October 25, 2022
Accepted:	October 28, 2022

Editor: Erik Hom

Reviewer(s): The reviewers have opted to remain anonymous.

Transaction Report:

DOI: <https://doi.org/10.1128/spectrum.01215-22>

July 15, 2022

Prof. Jian Lyu
Gansu Agricultural University
Anning District
Lanzhou
China

Re: Spectrum01215-22 (Changes in the microbial structure of the root soil and the yield of Chinese baby cabbage by chemical fertilizer reduction with bio-organic fertilizer application)

Dear Prof. Jian Lyu:

Thank you for submitting your manuscript to Microbiology Spectrum. Your manuscript has been reviewed now by two experts and their comments are attached. I concur with Reviewer #2's critiques, which point out major problems with your manuscript that I believe could warrant rejection. However, I wanted to give you an opportunity to address these. If you feel you can dramatically improve your manuscript to address the concerns raised, I would be willing to consider a revision. Note that I may send out your revised manuscript for re-review before making a final decision, and that this opportunity to revise does not guarantee acceptance.

When submitting the revised version of your paper, please provide (1) a summary of substantial changes you have made in your cover letter to me, (2) point-by-point responses to the issues raised by the reviewers as file type "Response to Reviewers," not in your cover letter, and (3) a PDF file that indicates the changes from the original submission (by highlighting or underlining the changes) as file type "Marked Up Manuscript - For Review Only". Please use this link to submit your revised manuscript - we strongly recommend that you submit your paper within the next 60 days or reach out to me. Detailed instructions on submitting your revised paper are below.

Link Not Available

Sincerely,

Erik Hom

Journals Department
Reviewer comments:

Reviewer #1 (Comments for the Author):

- Line 40 - correct the scientific name (subsp not ital.; small letter p in pekinensis)
- Line 41 - italicize *Brassica oleracea*; confirm if its *B. oleracea* or *B. rapa*
- Lines 91 to 94 - convert the bio-organic fertilizer amounts to rates (amount/area)
- Line 120-121 - does the bio-organic fertilizer also contain compost? What is the carrier for the microorganisms? If there is compost, do you have a nutrient analysis for the compost component of the bio-organic fertilizer?
- Line 122 - is this basic fertilizer or basal fertilizer?
- Lines 126-129 - indicate the rates of fertilizer application for conventional and each of the treatments
- (Line 135) Table 1 - are these the correct and actual fertilizer rates used? They seem too high than the usual rates applied. Is this the practice in the area? Are these the recommended rates for the bio-organic fertilizer? They also seem to be much higher than usual.
- Line 160 - 16S rRNA gene (add "gene")
- Line 171 - ddH₂O (subscript 2)
- Lines 199 to 200 - What was measured? Dry weight?
- Lines 331-332 - reference?
- Conclusion - answer to obj 5? Recommendation on optimal amount of supplementary fertilizer and basis for the recommendation?

Reviewer #2 (Comments for the Author):

The work reported in this MS describes the response of the rhizosphere microbiota to changes in the amount of chemical fertilizer used and the application of different amounts of bio-organic fertilizer. The authors show that reducing chemical fertilizer and addition of bio-organic fertilizer influences the rhizosphere bacterial and fungal diversity and results in changes in bacterial and fungal rhizosphere community composition. The authors also show that reducing chemical fertilizer and supplementing with bio-organic fertilizer increases cabbage yield. The results reported appear broadly consistent with similar studies performed in different plant models. The manuscript is missing important experimental design and method information that are important for interpreting the data. Below I describe my concerns with this manuscript.

MAJOR COMMENTS:

- 1) A major aim of this study is to determine how the application of bio-organic fertilizer influences the rhizosphere community of cabbage. Nowhere in the manuscript do the authors adequately describe the bio-organic fertilizer used, i.e. it is unclear what it is composed of, how it was created, or its biological properties. In line 120-122 the authors briefly say the bio-organic fertilizer has two "effective strains", *B. subtilis* and *P. stutzeri* (>20 million/g) and >40 organic matter. Given this description it is unclear if the fertilizer contains additional microbes and important information about the source/composition of the organic matter is not included. Since a major aim of this study is to document the microbial changes associated with bio-organic fertilizer application, the authors should provide more information about the bio-organic fertilizer input.
- 2) Study design and approach: In lines 139-140, the author describes the plants chosen for rhizosphere analysis as "30 plants with consistent growth were selected per plot". This description indicates that the authors used growth as a main factor to decide which plant rhizospheres would be characterized. This selection schema could bias the data generated. The authors should clearly describe how plant sampling was determined.
- 3) Yield : The authors do not adequately describe how they determined yield. In Lines: 198-201, the authors indicate that 30 plants were selected per plot and used to measure yield. They do not describe how/why these plants were chosen. It is unclear whether the plant selection process introduced bias into the yield determination. The authors should clearly describe the method used to determine yield, particularly the sampling methods.
- 4) Throughout the manuscript, the authors indicate where there is a statistical difference between treatments, but they do not describe whether these statistical differences are meaningful. For example, in figure 1, the authors suggest that T1 treatment results in a statistically significant increase in bacterial diversity. The data presented in figure 1A show that T1 rhizosphere Shannon index is ~0.05 higher than the chemical fertilizer only treatment-it is unclear how meaningful this difference is, despite its statistical significance and the authors do not discuss in detail these results. Where differences are indicated, the authors should discuss the relevance of the change to its biological/chemical function.

MINOR COMMENTS:

Throughout the MS the authors use rhizosphere and soil interchangeably. These are two distinct ecosystems and the authors should be consistent.

Line 34: you did not perform experiments with bio-organic fertilizer alone and therefore cannot conclude the combination is more efficacious than either treatment alone.

Line 302: remove "very". It is either statistically significant or not.

Line 401-402: You indicate that bio-organic fertilizer increased Proteobacteria and Firmicutes abundance. Please note that this change in abundance is likely due to the presence of Bacillus (Firmicutes) and Pseudomonas (Proteobacteria) in the bio-organic fertilizer. This should be discussed and any other microbes in the bio-organic fertilizer should be discussed.

Staff Comments:

Preparing Revision Guidelines

Please return the manuscript within 60 days; if you cannot complete the modification within this time period, please contact me. If you do not wish to modify the manuscript and prefer to submit it to another journal, please notify me of your decision immediately so that the manuscript may be formally withdrawn from consideration by Microbiology Spectrum.

The work reported in this MS describes the response of the rhizosphere microbiota to changes in the amount of chemical fertilizer used and the application of different amounts of bio-organic fertilizer. The authors show that reducing chemical fertilizer and addition of bio-organic fertilizer influences the rhizosphere bacterial and fungal diversity and results in changes in bacterial and fungal rhizosphere community composition. The authors also show that reducing chemical fertilizer and supplementing with bio-organic fertilizer increases cabbage yield. The results reported appear broadly consistent with similar studies performed in different plant models. The manuscript is missing important experimental design and method information that are important for interpreting the data. Given these deficiencies, I do not believe this MS meets the high technical quality and useful to community benchmark for publication in Microbiology Spectrum.

MAJOR COMMENTS:

- 1) A major aim of this study is to determine how the application of bio-organic fertilizer influences the rhizosphere community of cabbage. Nowhere in the manuscript do the authors adequately describe the bio-organic fertilizer used, *i.e.* it is unclear what it is composed of, how it was created, or its biological properties. In line 120-122 the authors briefly say the bio-organic fertilizer has two “effective strains”, *B. subtilis* and *P. stutzeri* (>20 million/g) and >40 organic matter. Given this description it is unclear if the fertilizer contains additional microbes and important information about the source/composition of the organic matter is not included. Since a major aim of this study is to document the microbial changes associated with bio-organic fertilizer application, the authors should provide more information about the bio-organic fertilizer input.
- 2) **Study design and approach:** In lines 139-140, the author describes the plants chosen for rhizosphere analysis as “30 plants with consistent growth were selected per plot”. This description indicates that the authors used growth as a main factor to decide which plant rhizospheres would be characterized. This selection schema could bias the data generated. The authors should clearly describe how plant sampling was determined.
- 3) **Yield :** The authors do not adequately describe how they determined yield. In Lines: 198-201, the authors indicate that 30 plants were selected per plot and used to measure yield. They do not describe how/why these plants were chosen. It is unclear whether the plant selection process introduced bias into the yield determination. The authors should clearly describe the method used to determine yield, particularly the sampling methods.
- 4) Throughout the manuscript, the authors indicate where there is a statistical difference between treatments, but they do not describe whether these statistical differences are meaningful. For example, in figure 1, the authors suggest that T1 treatment results in a statistically significant increase in bacterial diversity. The data presented in figure 1A show that T1 rhizosphere Shannon index is ~0.05 higher than the chemical fertilizer

only treatment—it is unclear how meaningful this difference is, despite its statistical significance and the authors do not discuss in detail these results. Where differences are indicated, the authors should discuss the relevance of the change to its biological/chemical function.

Throughout the MS the authors use rhizosphere and soil interchangeably. These are two distinct ecosystems and the authors should be consistent.

Line 34: you did not perform experiments with bio-organic fertilizer alone and therefore cannot conclude the combination is more efficacious than either treatment alone.

Line 302: remove “very”. It is either statistically significant or not.

Line 401-402: You indicate that bio-organic fertilizer increased Proteobacteria and Firmicutes abundance. Please note that this change in abundance is likely due to the presence of *Bacillus* (Firmicutes) and *Pseudomonas* (Proteobacteria) in the bio-organic fertilizer. This should be discussed and any other microbes in the bio-organic fertilizer should be discussed.

Response to Reviewers

Thank you for your comments for our manuscript titled “**Changes in the microbial structure of the root soil and the yield of Chinese baby cabbage by chemical fertilizer reduction with bio-organic fertilizer application (Spectrum01215-22)**”. In the first, we have carefully revised the manuscript according to your valuable comments and suggestions. Your comments and suggestions helped us improve the quality of our manuscript, which we hope will meet your expectation to be published in your journal. The revisions are highlighted by yellow in the manuscript. Our response to the reviewer’s comments are as follows:

Reviewer #1

Q1: *Line 40 - correct the scientific name (subsp not ital.; small letter p in pekinensis)*

Response: Thanks to your valuable suggestion. We have corrected the letter "P" to lowercase. **Line 39**

Q2: *Line 41 - italicize Brassica oleracea; confirm if its B. oleracea or B. rapa*

Response: Thank you for your kind suggestion. We confirmed that it was *B. oleracea* and italicized it. **Line 40**

Q3: *Lines 91 to 94 - convert the bio-organic fertilizer amounts to rates (amount/area)*

Response: Thank you for your helpful suggestion. We have converted the amount of bio-organic fertilizer into the rate, and the specific modifications are as follows:

“In the present study, we conducted field trials on Chinese baby cabbage under conventional fertilization, 30% chemical fertilizer reduction + 6,000 kg/ha bio-organic fertilizer, 30% chemical fertilizer reduction + 9,000 kg/ha bio-organic

fertilizer, 40% chemical fertilizer reduction + 6,000 kg/ha bio-organic fertilizer, and 40% chemical fertilizer reduction + 9,000 kg/ha bio-organic fertilizer.” Line 89-92

Q4: Line 120-121 - does the bio-organic fertilizer also contain compost? What is the carrier for the microorganisms? If there is compost, do you have a nutrient analysis for the compost component of the bio-organic fertilizer?

Response: Thanks to your valuable suggestion. After consulting with the company providing bio-organic fertilizer, we have obtained the main information of the bio-organic fertilizer, and they agreed to provide us with the preparation procedures of their bio-organic fertilizer. We have added its main preparation procedures to the manuscript, as follows:

“Bio-organic fertilizer was provided by Gansu Luneng Agricultural Science and Technology Co., Ltd., and was prepared as follows:

Bacillus subtilis and *Pseudomonas stutzeri* were provided by the Microbial Strain Conservation Center of Gansu Province and composed the microorganisms in the biological product. *B. subtilis* and *P. stutzeri* were added to the sterilized beef extract peptone medium (3.0 g beef extract, 10.0 g peptone, 5.0 g sodium chloride, 1000 ml water, 20.0 g agar; pH 7.4-7.6, 121 °C high-pressure steam sterilized for 20 min) in a 500 ml shaking flask, and cultured at 30 °C and 200 r / min for 24 h. The suspension of *B. subtilis* and *P. stutzeri* was then prepared into a compound bacterial solution at a 1:1 ratio. This complex bacterial solution was directly used to prepare biological products as described below.

The organic fertilizer used for biological products was composed of sheep manure and humic acid (formed from activated weathered coal) compost. Sheep manure and humic acid (4:1, w / w) were mixed evenly and stacked into strip ridges with a width of 1.5-5.0 m and height of 0.8-2.5 m for aerobic fermentation. The fermentation temperature was maintained between 40-60 °C, and the piles were turned over every 4-6 days until the end of fermentation. The content of organic matter, humic acid, and water in the fermented organic fertilizer were 40%, 25% and

30%, respectively, and contains 2.1% nitrogen, 0.7% P₂O₅, 1.2% K₂O and pH=6.8.

Bio-organic fertilizer was prepared by adding compound bacterial solution (inoculation amount: 20%) to the above fermented organic fertilizer and fermenting again for 5-6 days. After fermentation, the density of *B. subtilis* and *P. stutzeri* in the bio-organic fertilizer was 0.2×10^8 CFU/g.” **Line 104-127**

Q5: *Line 122 - is this basic fertilizer or basal fertilizer?*

Response: Thank you for your kind suggestion. In this experiment, we applied basal fertilizer at the early stage of crop growth. **Line 143**

Q6: *Lines 126-129 - indicate the rates of fertilizer application for conventional and each of the treatments*

Response: Thanks to your valuable suggestion. We have converted the amount of fertilizer applied to each treatment in the manuscript into the rate, and the specific modifications are as follows:

“The experimental design comprised conventional fertilizer (CK), 30% chemical fertilizer reduction + 6,000 kg/ha bio-organic fertilizer (T1), 30% chemical fertilizer reduction + 9,000 kg/ha bio-organic fertilizer (T2), 40% chemical fertilizer reduction + 6,000 kg/ha bio-organic fertilizer (T3), and 40% chemical fertilizer reduction + 9,000 kg/ha bio-organic fertilizer (T4).” **Line 147-150**

Q7: *(Line 135) Table 1 - are these the correct and actual fertilizer rates used? They seem too high than the usual rates applied. Is this the practice in the area? Are these the recommended rates for the bio-organic fertilizer? They also seem to be much higher than usual.*

Response: Thank you for your kind suggestion. Before the experiment, we did a lot

of investigations, and the traditional fertilizer application rate used in this experiment was also based on the investigation results. During the investigation, we found that the local soil is relatively infertile and farmers have a deep-rooted idea of increasing the yield by applying more chemical fertilizers, which leads to the phenomenon of blindly applying fertilizers and increasing the amounts of fertilizers. Therefore, we conducted this experiment to verify whether the low amount of chemical fertilizer combined with organic fertilizer can maintain or even increase the yield. In this experiment, we reduced chemical fertilizers by 30%-40% based on the traditional fertilizer application and applied 6000-9000 kg/ha of bio-organic fertilizers, and finally adjusted the ratio of nitrogen, phosphorus and potassium to achieve the appropriate amount of fertilizer required by the crop, thus avoiding fertilizer damage in vegetable fields, reducing soil pollution, improving soil microcosm and making land use sustainable. In addition, we converted the amount of chemical fertilizer into the amount of N, P₂O₅, and K₂O in order to see more clearly the amount of fertilizer applied to each treatment.

Table 1 Fertilization rates for each treatment

Treatments	Total amount of fertilization over the entire growth period (kg/ha)			
	N	P	K	Bio-organic fertilizer
CK	566.3	321.4	129.1	0.0
T1	348.3	166.6	196.9	6000
T2	348.3	166.6	196.9	9000
T3	298.5	142.8	168.7	6000
T4	298.5	142.8	168.7	9000

Q8: *Line 160 - 16S rRNA gene (add "gene")*

Response: Thank you for your helpful suggestion. We have added "gene" to the manuscript. **Line 182**

Q9: *Line 171 - ddH₂O (subscript 2)*

Response: Thank you for your kind suggestion. We have subscript "2" in ddH₂O.
Line 192

Q10: *Lines 199 to 200 - What was measured? Dry weight?*

Response: Thanks to your valuable suggestion. In this experiment, we used the equidistant sampling method to determine the yield. The yield is determined by the fresh weight of the plant. The specific determination methods are supplemented in the manuscript as follows:

“On September 16, 2019 and September 11, 2020, we used a five-point sampling method for yield determination. We artificially harvested in each sample point with an area of 1 m × 1 m, and a total of 30 plant samples were harvested for each treatment to obtain the fresh weight. The yield per hectare was given by $\text{yield} = (n * W)/30$, where W is the weight of 30 Chinese baby cabbages, and n is the number of Chinese baby cabbages per hectare.” Line 219-224

Q11: *Lines 331-332 - reference?*

Response: Thank you for your kind suggestion. We have supplemented the corresponding references in the manuscript.

“38. Gu S, Hu Q, Cheng Y, Bai L, Liu Z, Xiao W, Gong Z, Wu Y, Feng K, Deng Y. 2019. Application of organic fertilizer improves microbial community diversity and alters microbial network structure in tea (*Camellia sinensis*) plantation soils. *Soil and Tillage Research* 195:104356.” Line 354

Q12: *Conclusion - answer to obj 5? Recommendation on optimal amount of supplementary fertilizer and basis for the recommendation?*

Response: Thank you for your helpful suggestion. Our previous statement of obj 5 was inaccurate. I have corrected it to "5) recommend an optimal fertilization method

for the growth of Chinese baby cabbage." (Line100-101). In the conclusion, we also added the answer to goal 5. The details are as follows:

“Overall, the T2 fertilization formulation (30% chemical fertilizer reduction + 9,000 kg/ha bio-organic fertilizer) was more beneficial for soil microbial community structure regulation in the root soil of Chinese baby cabbages, which further promoted plant growth and development.” Line 494-497

Reviewer #2

Q1: *A major aim of this study is to determine how the application of bio-organic fertilizer influences the rhizosphere community of cabbage. Nowhere in the manuscript do the authors adequately describe the bio-organic fertilizer used, i.e. it is unclear what it is composed of, how it was created, or its biological properties. In line 120-122 the authors briefly say the bio-organic fertilizer has two "effective strains", *B. subtilis* and *P. stutzeri* (>20 million/g) and >40 organic matter. Given this description it is unclear if the fertilizer contains additional microbes and important information about the source/composition of the organic matter is not included. Since a major aim of this study is to document the microbial changes associated with bio-organic fertilizer application, the authors should provide more information about the bio-organic fertilizer input.*

Response: Thanks to your valuable suggestion. After we negotiated with the company providing bio-organic fertilizer, they agreed to provide us with their bio-organic fertilizer preparation procedures. We have added its main preparation procedures to the manuscript, as follows:

“Bio-organic fertilizer was provided by Gansu Luneng Agricultural Science and Technology Co., Ltd., and was prepared as follows:

Bacillus subtilis and *Pseudomonas stutzeri* were provided by the Microbial Strain Conservation Center of Gansu Province and composed the microorganisms in

the biological product. *B. subtilis* and *P. stutzeri* were added to the sterilized beef extract peptone medium (3.0 g beef extract, 10.0 g peptone, 5.0 g sodium chloride, 1000 ml water, 20.0 g agar; pH 7.4-7.6, 121 °C high-pressure steam sterilized for 20 min) in a 500 ml shaking flask, and cultured at 30 °C and 200 r / min for 24 h. The suspension of *B. subtilis* and *P. stutzeri* was then prepared into a compound bacterial solution at a 1:1 ratio. This complex bacterial solution was directly used to prepare biological products as described below.

The organic fertilizer used for biological products was composed of sheep manure and humic acid (formed from activated weathered coal) compost. Sheep manure and humic acid (4:1, w / w) were mixed evenly and stacked into strip ridges with a width of 1.5-5.0 m and height of 0.8-2.5 m for aerobic fermentation. The fermentation temperature was maintained between 40-60 °C, and the piles were turned over every 4-6 days until the end of fermentation. The content of organic matter, humic acid, and water in the fermented organic fertilizer were 40%, 25% and 30%, respectively, and contains 2.1% nitrogen, 0.7% P₂O₅, 1.2% K₂O and pH=6.8.

Bio-organic fertilizer was prepared by adding compound bacterial solution (inoculation amount: 20%) to the above fermented organic fertilizer and fermenting again for 5-6 days. After fermentation, the density of *B. subtilis* and *P. stutzeri* in the bio-organic fertilizer was 0.2×10^8 CFU/g.” **Line 104-127**

Q2: *Study design and approach: In lines 139-140, the author describes the plants chosen for rhizosphere analysis as "30 plants with consistent growth were selected per plot". This description indicates that the authors used growth as a main factor to decide which plant rhizospheres would be characterized. This selection schema could bias the data generated. The authors should clearly describe how plant sampling was determined.*

Response: Thank you for your helpful suggestion. We have not clearly stated the sampling method before. We have now corrected it to " Each plot was sampled with a 5-point sampling method, with 6 plants selected for each sample point, and a total of

30 plant roots selected for soil sampling. Line 158-161" in the manuscript.

Q3: *Yield: The authors do not adequately describe how they determined yield. In Lines: 198-201, the authors indicate that 30 plants were selected per plot and used to measure yield. They do not describe how/why these plants were chosen. It is unclear whether the plant selection process introduced bias into the yield determination. The authors should clearly describe the method used to determine yield, particularly the sampling methods.*

Response: Thank you for your helpful suggestion. We added more detailed methods for determining yield in the manuscript, as follows:

“On September 16, 2019 and September 11, 2020, we used a five-point sampling method for yield determination. We artificially harvested in each sample point with an area of 1 m × 1 m, and a total of 30 plant samples were harvested for each treatment to obtain the fresh weight. The yield per hectare was given by $\text{yield} = (n * W)/30$, where W is the weight of 30 Chinese baby cabbages, and n is the number of Chinese baby cabbages per hectare.” Line 219-224

Q4: *Throughout the manuscript, the authors indicate where there is a statistical difference between treatments, but they do not describe whether these statistical differences are meaningful. For example, in figure 1, the authors suggest that T1 treatment results in a statistically significant increase in bacterial diversity. The data presented in figure 1A show that T1 rhizosphere Shannon index is ~0.05 higher than the chemical fertilizer only treatment-it is unclear how meaningful this difference is, despite its statistical significance and the authors do not discuss in detail these results. Where differences are indicated, the authors should discuss the relevance of the change to its biological/chemical function.*

Response: Thank you for your kind suggestion. After carefully reviewing a large number of references, we found that in many cases, the Shannon index of soil bacteria

can be significantly different in a small range, such as the following three references:

1. Chen, Chen, et al. "Microbial communities of an arable soil treated for 8 years with organic and inorganic fertilizers." *Biology and Fertility of Soils* 52.4 (2016): 455-467.
2. Han, Jianqiao, Yunyun Dong, and Man Zhang. "Chemical fertilizer reduction with organic fertilizer effectively improve soil fertility and microbial community from newly cultivated land in the Loess Plateau of China." *Applied Soil Ecology* 165 (2021): 103966.
3. Ji, Lingfei, et al. "Effects of organic substitution for synthetic N fertilizer on soil bacterial diversity and community composition: A 10-year field trial in a tea plantation." *Agriculture, Ecosystems & Environment* 268 (2018): 124-132."

In the discussion part, we supplemented the detailed discussion on this difference, as follows:

“Our results showed that compared with conventional fertilization, T1 and T3 significantly increased bacterial diversity while T4 significantly increased bacterial richness (Fig. 1A-C). Previous studies have also shown that under chemical fertilizer reduction conditions, organic supplementation can significantly alter the soil bacterial community structure, while also significantly increasing bacterial species richness, chao1, and Shannon index values (46, 47). This may be because organic fertilizers can not only provide greater diversity of microbial active substrates than inorganic fertilizers, but also directly introduce naturally occurring microorganisms into the soil (48). In our experiments, organic fertilizers were supplemented with additional beneficial microorganisms, which beneficially increased the diversity and richness of the bacterial community. In addition, the novelty of our study was to compare the effects of different ratios of chemical fertilizers and bio-organic fertilizers on the diversity and richness of microbial communities. From our results, it is evident that different ratios of chemical fertilizers and bio-organic fertilizers have varying effects on the diversity and richness of bacterial communities. Therefore, optimized fertilizer ratios should be selected during the cultivation of Chinese baby cabbage.” **Line**

Q5: *Throughout the MS the authors use rhizosphere and soil interchangeably. These are two distinct ecosystems and the authors should be consistent.*

Response: Thank you for your kind suggestion. We have unified it as "soil" in the manuscript.

Q6: *Line 34: you did not perform experiments with bio-organic fertilizer alone and therefore cannot conclude the combination is more efficacious than either treatment alone.*

Response: Thank you for your helpful suggestion. Our statement here is incorrect and has been corrected to " Bio-organic + chemical fertilizer was more effective than chemical fertilizer alone ". **Line 34**

Q7: *Line 302: remove "very". It is either statistically significant or not.*

Response: Thank you for your kind suggestion. We have removed "very" from the manuscript. **Line 323**

Q8: *Line 401-402: You indicate that bio-organic fertilizer increased Proteobacteria and Firmicutes abundance. Please note that this change in abundance is likely due to the presence of Bacillus (Firmicutes) and Pseudomonas (Proteobacteria) in the bio-organic fertilizer. This should be discussed and any other microbes in the bio-organic fertilizer should be discussed.*

Response: Thanks to your valuable suggestion. We have supplemented the discussion on Proteobacteria and Firmicutes in the manuscript, as follows:

“In this study, bio-organic fertilizer treatment also increased *Proteobacteria* and

Firmicutes abundance. This change in abundance may be due to the presence of *B. subtilis* (*Firmicutes*) and *P. stutzeri* (*Proteobacteria*) in the bio-organic fertilizers used in this study. *Bacillus* spp. and *Pseudomonas* spp. are widely used bacterial and fungal biocontrol agents, and are used to control soil-borne plant diseases (67, 68). Cooperation between *Bacillus* spp. and *Pseudomonas* spp. can induce positive interactions and generate multispecies biofilms at the root-microbiome interface, which can potentially trigger microbial root colonization and resultant plant disease resistance (69-72). In addition, in this study, only *B. subtilis* and *P. stutzeri* were included in the bio- organic fertilizer formula. The enrichment of other potentially beneficial microorganisms in the root soil of Chinese baby cabbages may be due to the inhibition of soil-borne pathogens and rebalancing of the microbial community structure. Therefore, the strategy of reducing chemical fertilizer and applying bio-organic fertilizer for changing plant soil microbial community composition has potential for controlling soil-borne diseases and improving plant growth.” Line

434-449

October 18, 2022

Prof. Jian Lyu
Gansu Agricultural University
Anning District
Lanzhou
China

Re: Spectrum01215-22R1 (Changes in the microbial structure of the root soil and the yield of Chinese baby cabbage by chemical fertilizer reduction with bio-organic fertilizer application)

Dear Prof. Jian Lyu:

Thank you for submitting your manuscript to Microbiology Spectrum. I apologize for the long delay in the review process. Given Reviewer #1's comments and my own assessment of your responses/revisions, I am going to conditionally accept your manuscript *provided* you better address the original critique of Reviewer #2:

Q2: Study design and approach: In lines 139-140, the author describes the plants chosen for rhizosphere analysis as "30 plants with consistent growth were selected per plot". This description indicates that the authors used growth as a main factor to decide which plant rhizospheres would be characterized. This selection schema could bias the data generated. The authors should clearly describe how plant sampling was determined.

Response: Thank you for your helpful suggestion. We have not clearly stated the sampling method before. We have now corrected it to " Each plot was sampled with a 5-point sampling method, with 6 plants selected for each sample point, and a total of 30 plant roots selected for soil sampling. Line 158-161" in the manuscript.

I believe what Reviewer #2 was originally asking was also to provide details for HOW you selected the 6 different plants in the 5-point sampling you did. Was this random? Did you purposely choose plants that were the same size? Etc.

If you could kindly address this in your revision and summarize what you have done in your cover letter to me in response to this, I can go ahead and accept if you provide the additional information.

Link Not Available

Sincerely,

Erik Hom

Journals Department
Reviewer comments:

Staff Comments:

Preparing Revision Guidelines

Please return the manuscript within 60 days; if you cannot complete the modification within this time period, please contact me. If you do not wish to modify the manuscript and prefer to submit it to another journal, please notify me of your decision immediately so that the manuscript may be formally withdrawn from consideration by Microbiology Spectrum.

Response to Reviewers

Thank you for your comments for our manuscript titled “**Changes in the microbial structure of the root soil and the yield of Chinese baby cabbage by chemical fertilizer reduction with bio-organic fertilizer application (Spectrum01215-22)**”. In the first, we have carefully revised the manuscript according to your valuable comments and suggestions. Your comments and suggestions helped us improve the quality of our manuscript, which we hope will meet your expectation to be published in your journal. The revisions are highlighted by yellow in the manuscript. Our response to the reviewer’s comments are as follows:

Reviewer #1

Q1: *Line 40 - correct the scientific name (subsp not ital.; small letter p in pekinensis)*

Response: Thanks to your valuable suggestion. We have corrected the letter "P" to lowercase. **Line 39**

Q2: *Line 41 - italicize Brassica oleracea; confirm if its B. oleracea or B. rapa*

Response: Thank you for your kind suggestion. We confirmed that it was *B. oleracea* and italicized it. **Line 40**

Q3: *Lines 91 to 94 - convert the bio-organic fertilizer amounts to rates (amount/area)*

Response: Thank you for your helpful suggestion. We have converted the amount of bio-organic fertilizer into the rate, and the specific modifications are as follows:

“In the present study, we conducted field trials on Chinese baby cabbage under conventional fertilization, 30% chemical fertilizer reduction + 6,000 kg/ha bio-organic fertilizer, 30% chemical fertilizer reduction + 9,000 kg/ha bio-organic

fertilizer, 40% chemical fertilizer reduction + 6,000 kg/ha bio-organic fertilizer, and 40% chemical fertilizer reduction + 9,000 kg/ha bio-organic fertilizer.” Line 89-92

Q4: Line 120-121 - does the bio-organic fertilizer also contain compost? What is the carrier for the microorganisms? If there is compost, do you have a nutrient analysis for the compost component of the bio-organic fertilizer?

Response: Thanks to your valuable suggestion. After consulting with the company providing bio-organic fertilizer, we have obtained the main information of the bio-organic fertilizer, and they agreed to provide us with the preparation procedures of their bio-organic fertilizer. We have added its main preparation procedures to the manuscript, as follows:

“Bio-organic fertilizer was provided by Gansu Luneng Agricultural Science and Technology Co., Ltd., and was prepared as follows:

Bacillus subtilis and *Pseudomonas stutzeri* were provided by the Microbial Strain Conservation Center of Gansu Province and composed the microorganisms in the biological product. *B. subtilis* and *P. stutzeri* were added to the sterilized beef extract peptone medium (3.0 g beef extract, 10.0 g peptone, 5.0 g sodium chloride, 1000 ml water, 20.0 g agar; pH 7.4-7.6, 121 °C high-pressure steam sterilized for 20 min) in a 500 ml shaking flask, and cultured at 30 °C and 200 r / min for 24 h. The suspension of *B. subtilis* and *P. stutzeri* was then prepared into a compound bacterial solution at a 1:1 ratio. This complex bacterial solution was directly used to prepare biological products as described below.

The organic fertilizer used for biological products was composed of sheep manure and humic acid (formed from activated weathered coal) compost. Sheep manure and humic acid (4:1, w / w) were mixed evenly and stacked into strip ridges with a width of 1.5-5.0 m and height of 0.8-2.5 m for aerobic fermentation. The fermentation temperature was maintained between 40-60 °C, and the piles were turned over every 4-6 days until the end of fermentation. The content of organic matter, humic acid, and water in the fermented organic fertilizer were 40%, 25% and

30%, respectively, and contains 2.1% nitrogen, 0.7% P₂O₅, 1.2% K₂O and pH=6.8.

Bio-organic fertilizer was prepared by adding compound bacterial solution (inoculation amount: 20%) to the above fermented organic fertilizer and fermenting again for 5-6 days. After fermentation, the density of *B. subtilis* and *P. stutzeri* in the bio-organic fertilizer was 0.2×10^8 CFU/g.” **Line 104-127**

Q5: *Line 122 - is this basic fertilizer or basal fertilizer?*

Response: Thank you for your kind suggestion. In this experiment, we applied basal fertilizer at the early stage of crop growth. **Line 143**

Q6: *Lines 126-129 - indicate the rates of fertilizer application for conventional and each of the treatments*

Response: Thanks to your valuable suggestion. We have converted the amount of fertilizer applied to each treatment in the manuscript into the rate, and the specific modifications are as follows:

“The experimental design comprised conventional fertilizer (CK), 30% chemical fertilizer reduction + 6,000 kg/ha bio-organic fertilizer (T1), 30% chemical fertilizer reduction + 9,000 kg/ha bio-organic fertilizer (T2), 40% chemical fertilizer reduction + 6,000 kg/ha bio-organic fertilizer (T3), and 40% chemical fertilizer reduction + 9,000 kg/ha bio-organic fertilizer (T4).” **Line 147-150**

Q7: *(Line 135) Table 1 - are these the correct and actual fertilizer rates used? They seem too high than the usual rates applied. Is this the practice in the area? Are these the recommended rates for the bio-organic fertilizer? They also seem to be much higher than usual.*

Response: Thank you for your kind suggestion. Before the experiment, we did a lot

of investigations, and the traditional fertilizer application rate used in this experiment was also based on the investigation results. During the investigation, we found that the local soil is relatively infertile and farmers have a deep-rooted idea of increasing the yield by applying more chemical fertilizers, which leads to the phenomenon of blindly applying fertilizers and increasing the amounts of fertilizers. Therefore, we conducted this experiment to verify whether the low amount of chemical fertilizer combined with organic fertilizer can maintain or even increase the yield. In this experiment, we reduced chemical fertilizers by 30%-40% based on the traditional fertilizer application and applied 6000-9000 kg/ha of bio-organic fertilizers, and finally adjusted the ratio of nitrogen, phosphorus and potassium to achieve the appropriate amount of fertilizer required by the crop, thus avoiding fertilizer damage in vegetable fields, reducing soil pollution, improving soil microcosm and making land use sustainable. In addition, we converted the amount of chemical fertilizer into the amount of N, P₂O₅, and K₂O in order to see more clearly the amount of fertilizer applied to each treatment.

Table 1 Fertilization rates for each treatment

Treatments	Total amount of fertilization over the entire growth period (kg/ha)			
	N	P	K	Bio-organic fertilizer
CK	566.3	321.4	129.1	0.0
T1	348.3	166.6	196.9	6000
T2	348.3	166.6	196.9	9000
T3	298.5	142.8	168.7	6000
T4	298.5	142.8	168.7	9000

Q8: *Line 160 - 16S rRNA gene (add "gene")*

Response: Thank you for your helpful suggestion. We have added "gene" to the manuscript. **Line 182**

Q9: *Line 171 - ddH₂O (subscript 2)*

Response: Thank you for your kind suggestion. We have subscript "2" in ddH₂O.
Line 192

Q10: *Lines 199 to 200 - What was measured? Dry weight?*

Response: Thanks to your valuable suggestion. In this experiment, we used the equidistant sampling method to determine the yield. The yield is determined by the fresh weight of the plant. The specific determination methods are supplemented in the manuscript as follows:

“On September 16, 2019 and September 11, 2020, we used a five-point sampling method for yield determination. We artificially harvested in each sample point with an area of 1 m × 1 m, and a total of 30 plant samples were harvested for each treatment to obtain the fresh weight. The yield per hectare was given by $\text{yield} = (n * W)/30$, where W is the weight of 30 Chinese baby cabbages, and n is the number of Chinese baby cabbages per hectare.” Line 219-224

Q11: *Lines 331-332 - reference?*

Response: Thank you for your kind suggestion. We have supplemented the corresponding references in the manuscript.

“38. Gu S, Hu Q, Cheng Y, Bai L, Liu Z, Xiao W, Gong Z, Wu Y, Feng K, Deng Y. 2019. Application of organic fertilizer improves microbial community diversity and alters microbial network structure in tea (*Camellia sinensis*) plantation soils. *Soil and Tillage Research* 195:104356.” Line 354

Q12: *Conclusion - answer to obj 5? Recommendation on optimal amount of supplementary fertilizer and basis for the recommendation?*

Response: Thank you for your helpful suggestion. Our previous statement of obj 5 was inaccurate. I have corrected it to "5) recommend an optimal fertilization method

for the growth of Chinese baby cabbage." (Line100-101). In the conclusion, we also added the answer to goal 5. The details are as follows:

“Overall, the T2 fertilization formulation (30% chemical fertilizer reduction + 9,000 kg/ha bio-organic fertilizer) was more beneficial for soil microbial community structure regulation in the root soil of Chinese baby cabbages, which further promoted plant growth and development.” Line 494-497

Reviewer #2

Q1: *A major aim of this study is to determine how the application of bio-organic fertilizer influences the rhizosphere community of cabbage. Nowhere in the manuscript do the authors adequately describe the bio-organic fertilizer used, i.e. it is unclear what it is composed of, how it was created, or its biological properties. In line 120-122 the authors briefly say the bio-organic fertilizer has two "effective strains", *B. subtilis* and *P. stutzeri* (>20 million/g) and >40 organic matter. Given this description it is unclear if the fertilizer contains additional microbes and important information about the source/composition of the organic matter is not included. Since a major aim of this study is to document the microbial changes associated with bio-organic fertilizer application, the authors should provide more information about the bio-organic fertilizer input.*

Response: Thanks to your valuable suggestion. After we negotiated with the company providing bio-organic fertilizer, they agreed to provide us with their bio-organic fertilizer preparation procedures. We have added its main preparation procedures to the manuscript, as follows:

“Bio-organic fertilizer was provided by Gansu Luneng Agricultural Science and Technology Co., Ltd., and was prepared as follows:

Bacillus subtilis and *Pseudomonas stutzeri* were provided by the Microbial Strain Conservation Center of Gansu Province and composed the microorganisms in

the biological product. *B. subtilis* and *P. stutzeri* were added to the sterilized beef extract peptone medium (3.0 g beef extract, 10.0 g peptone, 5.0 g sodium chloride, 1000 ml water, 20.0 g agar; pH 7.4-7.6, 121 °C high-pressure steam sterilized for 20 min) in a 500 ml shaking flask, and cultured at 30 °C and 200 r / min for 24 h. The suspension of *B. subtilis* and *P. stutzeri* was then prepared into a compound bacterial solution at a 1:1 ratio. This complex bacterial solution was directly used to prepare biological products as described below.

The organic fertilizer used for biological products was composed of sheep manure and humic acid (formed from activated weathered coal) compost. Sheep manure and humic acid (4:1, w / w) were mixed evenly and stacked into strip ridges with a width of 1.5-5.0 m and height of 0.8-2.5 m for aerobic fermentation. The fermentation temperature was maintained between 40-60 °C, and the piles were turned over every 4-6 days until the end of fermentation. The content of organic matter, humic acid, and water in the fermented organic fertilizer were 40%, 25% and 30%, respectively, and contains 2.1% nitrogen, 0.7% P₂O₅, 1.2% K₂O and pH=6.8.

Bio-organic fertilizer was prepared by adding compound bacterial solution (inoculation amount: 20%) to the above fermented organic fertilizer and fermenting again for 5-6 days. After fermentation, the density of *B. subtilis* and *P. stutzeri* in the bio-organic fertilizer was 0.2×10^8 CFU/g.” **Line 104-127**

Q2: *Study design and approach: In lines 139-140, the author describes the plants chosen for rhizosphere analysis as "30 plants with consistent growth were selected per plot". This description indicates that the authors used growth as a main factor to decide which plant rhizospheres would be characterized. This selection schema could bias the data generated. The authors should clearly describe how plant sampling was determined.*

Response: Thank you for your helpful suggestion. We have not clearly stated the sampling method before. We have now corrected it to " Each plot was sampled with a 5-point sampling method, with 6 plants selected for each sample point, and a total of

30 plant roots selected for soil sampling. Line 158-161" in the manuscript.

Q3: *Yield: The authors do not adequately describe how they determined yield. In Lines: 198-201, the authors indicate that 30 plants were selected per plot and used to measure yield. They do not describe how/why these plants were chosen. It is unclear whether the plant selection process introduced bias into the yield determination. The authors should clearly describe the method used to determine yield, particularly the sampling methods.*

Response: Thank you for your helpful suggestion. We added more detailed methods for determining yield in the manuscript, as follows:

“On September 16, 2019 and September 11, 2020, we used a five-point sampling method for yield determination. We artificially harvested in each sample point with an area of 1 m × 1 m, and a total of 30 plant samples were harvested for each treatment to obtain the fresh weight. The yield per hectare was given by $\text{yield} = (n * W)/30$, where W is the weight of 30 Chinese baby cabbages, and n is the number of Chinese baby cabbages per hectare.” Line 219-224

Q4: *Throughout the manuscript, the authors indicate where there is a statistical difference between treatments, but they do not describe whether these statistical differences are meaningful. For example, in figure 1, the authors suggest that T1 treatment results in a statistically significant increase in bacterial diversity. The data presented in figure 1A show that T1 rhizosphere Shannon index is ~0.05 higher than the chemical fertilizer only treatment-it is unclear how meaningful this difference is, despite its statistical significance and the authors do not discuss in detail these results. Where differences are indicated, the authors should discuss the relevance of the change to its biological/chemical function.*

Response: Thank you for your kind suggestion. After carefully reviewing a large number of references, we found that in many cases, the Shannon index of soil bacteria

can be significantly different in a small range, such as the following three references:

1. Chen, Chen, et al. "Microbial communities of an arable soil treated for 8 years with organic and inorganic fertilizers." *Biology and Fertility of Soils* 52.4 (2016): 455-467.

2. Han, Jianqiao, Yunyun Dong, and Man Zhang. "Chemical fertilizer reduction with organic fertilizer effectively improve soil fertility and microbial community from newly cultivated land in the Loess Plateau of China." *Applied Soil Ecology* 165 (2021): 103966.

3. Ji, Lingfei, et al. "Effects of organic substitution for synthetic N fertilizer on soil bacterial diversity and community composition: A 10-year field trial in a tea plantation." *Agriculture, Ecosystems & Environment* 268 (2018): 124-132."

In the discussion part, we supplemented the detailed discussion on this difference, as follows:

"Our results showed that compared with conventional fertilization, T1 and T3 significantly increased bacterial diversity while T4 significantly increased bacterial richness (Fig. 1A-C). Previous studies have also shown that under chemical fertilizer reduction conditions, organic supplementation can significantly alter the soil bacterial community structure, while also significantly increasing bacterial species richness, chao1, and Shannon index values (46, 47). This may be because organic fertilizers can not only provide greater diversity of microbial active substrates than inorganic fertilizers, but also directly introduce naturally occurring microorganisms into the soil (48). In our experiments, organic fertilizers were supplemented with additional beneficial microorganisms, which beneficially increased the diversity and richness of the bacterial community. In addition, the novelty of our study was to compare the effects of different ratios of chemical fertilizers and bio-organic fertilizers on the diversity and richness of microbial communities. From our results, it is evident that different ratios of chemical fertilizers and bio-organic fertilizers have varying effects on the diversity and richness of bacterial communities. Therefore, optimized fertilizer ratios should be selected during the cultivation of Chinese baby cabbage." Line

Q5: *Throughout the MS the authors use rhizosphere and soil interchangeably. These are two distinct ecosystems and the authors should be consistent.*

Response: Thank you for your kind suggestion. We have unified it as "soil" in the manuscript.

Q6: *Line 34: you did not perform experiments with bio-organic fertilizer alone and therefore cannot conclude the combination is more efficacious than either treatment alone.*

Response: Thank you for your helpful suggestion. Our statement here is incorrect and has been corrected to " Bio-organic + chemical fertilizer was more effective than chemical fertilizer alone ". **Line 34**

Q7: *Line 302: remove "very". It is either statistically significant or not.*

Response: Thank you for your kind suggestion. We have removed "very" from the manuscript. **Line 323**

Q8: *Line 401-402: You indicate that bio-organic fertilizer increased Proteobacteria and Firmicutes abundance. Please note that this change in abundance is likely due to the presence of Bacillus (Firmicutes) and Pseudomonas (Proteobacteria) in the bio-organic fertilizer. This should be discussed and any other microbes in the bio-organic fertilizer should be discussed.*

Response: Thanks to your valuable suggestion. We have supplemented the discussion on Proteobacteria and Firmicutes in the manuscript, as follows:

“In this study, bio-organic fertilizer treatment also increased *Proteobacteria* and

Firmicutes abundance. This change in abundance may be due to the presence of *B. subtilis* (*Firmicutes*) and *P. stutzeri* (*Proteobacteria*) in the bio-organic fertilizers used in this study. *Bacillus* spp. and *Pseudomonas* spp. are widely used bacterial and fungal biocontrol agents, and are used to control soil-borne plant diseases (67, 68). Cooperation between *Bacillus* spp. and *Pseudomonas* spp. can induce positive interactions and generate multispecies biofilms at the root-microbiome interface, which can potentially trigger microbial root colonization and resultant plant disease resistance (69-72). In addition, in this study, only *B. subtilis* and *P. stutzeri* were included in the bio- organic fertilizer formula. The enrichment of other potentially beneficial microorganisms in the root soil of Chinese baby cabbages may be due to the inhibition of soil-borne pathogens and rebalancing of the microbial community structure. Therefore, the strategy of reducing chemical fertilizer and applying bio-organic fertilizer for changing plant soil microbial community composition has potential for controlling soil-borne diseases and improving plant growth.” Line

434-449

October 28, 2022

Prof. Jian Lyu
Gansu Agricultural University
Anning District
Lanzhou
China

Re: Spectrum01215-22R2 (Changes in the microbial structure of the root soil and the yield of Chinese baby cabbage by chemical fertilizer reduction with bio-organic fertilizer application)

Dear Prof. Jian Lyu:

Thank you for your edits. I am going to accept your manuscript BUT want you to make the following edit before publication; I am going to request that you modify the additional text as follows:

"A 5-point random sampling method was adopted for each plot of land with each sampling point having an area of 1 m × 1 m {REFERENCES}. Six plants in total were collected at each sampling point for a total of 30 plant roots for soil sampling." (Lines 158-161)

Please replace {REFERENCES} with the 4 references you cited in your cover letter to me of prior studies that used this 5-point method and update your reference list to reflect the additional references. This will be helpful to readers, so they know you didn't just make up a new method but applied an already existing one.

I am forwarding this version of the manuscript to the ASM Journals Department for publication, but please go ahead and send them a version with the revised text if you can; you can refer to my decision letter to alert them of this very minor revised manuscript. You will be notified when your proofs are ready to be viewed.

Sincerely,

Erik Hom
Editor, Microbiology Spectrum
